# Association of clinical and genetic risk factors with management of dyslipidaemia: analysis of repeated cross-sectional studies in the general population of Lausanne, Switzerland

Valeriya Chekanova,[1,2] Nazanin Abolhassani,[2,3] Julien Vaucher,[2]
Pedro Marques-Vidal  [2]

[1]National Medical Research Center of Cardiology, Moscow, Russian Federation
[2]Department of Medicine, Internal Medicine, Lausanne University Hospital and University of Lausanne, Lausanne, Switzerland
[3]Institute of Primary Health Care, Bern, Switzerland

**Correspondence to**
Professor Pedro Marques-Vidal; Pedro-Manuel.Marques-Vidal@chuv.ch

## ABSTRACT

**Objectives** To assess the importance of clinical and genetic factors in management of dyslipidaemia in the general population.

**Design** Repeated cross-sectional studies (2003–2006; 2009–2012 and 2014–2017) from a population-based cohort.

**Setting** Single centre in Lausanne, Switzerland.

**Participants** 617 (42.6% women, mean±SD: 61.6±8.5 years), 844 (48.5% women, 64.5±8.8 years) and 798 (50.3% women, 68.1±9.2) participants of the baseline, first and second follow-ups receiving any type of lipid-lowering drug. Participants were excluded if they had missing information regarding lipid levels, covariates or genetic data.

**Primary and secondary outcome measures** Management of dyslipidaemia was assessed according to European or Swiss guidelines. Genetic risk scores (GRSs) for lipid levels were computed based on the existing literature.

**Results** Prevalence of adequately controlled dyslipidaemia was 52%, 45% and 46% at baseline, first and second follow-ups, respectively. On multivariable analysis, when compared with intermediate or low-risk individuals, participants at very high cardiovascular risk had an OR for dyslipidaemia control of 0.11 (95% CI: 0.06 to 0.18), 0.12 (0.08 to 0.19) and 0.38 (0.25 to 0.59) at baseline, first and second follow-ups, respectively. Use of newer generation or higher potency statins was associated with better control: OR of 1.90 (1.18 to 3.05) and 3.62 (1.65 to 7.92) for second and third generations compared with first in the first follow-up, with the corresponding values in the second follow-up being 1.90 (1.08 to 3.36) and 2.18 (1.05 to 4.51). No differences in GRSs were found between controlled and inadequately controlled subjects. Similar findings were obtained using Swiss guidelines.

**Conclusion** Management of dyslipidaemia is suboptimal in Switzerland. The effectiveness of high potency statins is hampered by low posology. The use of GRSs in the management of dyslipidaemia is not recommended.

## STRENGTHS AND LIMITATIONS OF THIS STUDY

⇒ Multiple cross-sectional studies conducted in a population-based cohort.
⇒ Three different genetic risk scores and 51 single single nucleotide polymorphisms for lipids were tested.
⇒ Two criteria to define and treat dyslipidaemia were applied.
⇒ Lack of consensus regarding diagnosis and management of dyslipidaemia; results cannot be extrapolated to other settings and populations.
⇒ Results based on a single population and hence not forcefully generalisable to other settings and populations.

## INTRODUCTION

Adequate management of dyslipidaemia [high Low density lipoprotein (LDL)-cholesterol levels] translates into a reduction in fatal and non-fatal cardiovascular disease (CVD),[1 2] and guidelines for the management of dyslipidaemia have been issued by the European Society of Cardiology (ESC) and the European Atherosclerosis Society (EAS).[3] Potent hypolipidaemic drugs are available, allowing a considerable reduction in LDL-cholesterol levels.[3] Still, management of dyslipidaemia is suboptimal, with a significant percentage of treated patients not reaching target levels.[4] Likely contributing factors are inadequate perception of risk by physicians,[5] low compliance by patients[6] or use of lesser potent drugs.[4] It has also been suggested that the efficacy of statins, the main hypolipidaemic drugs used, could be modulated by the genetic background of the patients.[7 8] A recent review suggested that several single nucleotide polymorphisms (SNPs) could be associated with a reduction in the efficacy of statin treatment.[8] Still, the effect of genetic

markers on the management of dyslipidaemia in the general population has seldom been established.

Thus, we aimed to assess the importance of clinical and genetic factors in the management of dyslipidaemia using data from a population-based cohort.

## METHODS

### Study population

The CoLaus|PsyCoLaus (www.colaus-psycolaus.ch) is a prospective cohort study following every 5 years a sample of the inhabitants of the city of Lausanne (Switzerland, population 137810 in 2017), aged 35–75 years at baseline.[9] In the present study, data from the baseline (2003–2006), the first (2009–2012) and the second (2014–2017) follow-ups were used.

### Inclusion and exclusion criteria

Participants were eligible if they received any type of lipid-lowering drug. Participants were initially excluded if they had missing information regarding lipid levels, covariates, or genetic data.

### Lipid-lowering treatment and control of dyslipidaemia

At each survey, participants reported which drugs they were taking. Based on the Anatomical Therapeutic Chemical classification system of the WHO, participants were considered as being treated for dyslipidaemia if they were taking one drug coded C10 ('lipid modifying agents'). Lipid-lowering drugs were further classified into statins, fibrates and other lipid-lowering drugs. For statins, a further classification regarding the generation and potency was performed in the first and second follow-ups (online supplemental table 1). Such classification could not be achieved in the baseline survey due to limited coding. Two approaches regarding statin potency were conducted: (1) not taking into account and (2) taking into account posology as defined by US guidelines.[10] This last approach is similar to another study conducted in Poland.[11]

As there is no consensus regarding CVD risk assessment in Switzerland, two approaches were applied. The first approach used the ESC/EAS guidelines[3] by applying the SCORE equation recalibrated for Switzerland[12] (online supplemental table 2). Three CVD categories were defined: very high, high and other. The second approach used the Swiss Group for Lipids and Atherosclerosis (GSLA) criteria[13] (online supplemental table 3). Depending on the risk category, the threshold to define adequate control changed (online supplemental tables 2 and 3).

### Genetic analysis and genetic scores

Genome-wide genotyping was performed using the Affymetrix 500K SNP array. Subjects were excluded from the analysis in case of inconsistency between sex and genetic data, a genotype call rate of <90%, or inconsistencies of genotyping results in duplicate samples. Quality control for SNPs was performed using the following criteria: monomorphic (or with minor allele frequency <1%), call rates <90%, deviation from the Hardy-Weinberg equilibrium ($p<1\times10^{-6}$). Phased haplotypes were generated using SHAPEIT2.[14] Imputation was performed using minimac3 and the Haplotype Reference Consortium V.r1.1. Fifty-one SNPs associated with lipid-lowering drug efficiency were extracted (online supplemental table 4) from a previous review.[8] Genetic risk scores (GRSs) for total, LDL-cholesterol and HDL-cholesterol were computed using 223 SNPs overall as suggested previously.[15] Briefly, the GRSs were calculated with each SNP being weighted by its relative effect size (β coefficient) obtained from the literature (online supplemental table 5).

### Other covariates

Sociodemographic and lifestyle data were collected by questionnaire and included gender, age, educational level (low/middle/high), marital status (alone/couple), personal and family history of CVD, family history of dyslipidaemia, smoking (never/former/current) and alcohol consumption (yes/no). Number of other drugs (including or excluding non-prescribed, over-the-counter drugs) were considered as a proxy for the number of comorbidities.

Body weight and height were measured with participants barefoot and in light indoor clothes. Body weight was measured in kilograms to the nearest 100 g using a Seca scale (Hamburg, Germany). Height was measured to the nearest 5 mm using a Seca (Hamburg, Germany) height gauge. Body mass index (BMI) was calculated and categorised into normal ($<25\,\mathrm{kg/m^2}$), overweight ($25 \leq BMI < 30\,\mathrm{kg/m^2}$) and obese ($BMI \geq 30\,\mathrm{kg/m^2}$).

Blood pressure (BP) was measured using an Omron HEM-907 automated oscillometric sphygmomanometer after at least a 10-min rest in a seated position, and the average of the last two measurements was used. Hypertension was defined by a systolic blood pressure ≥140 mm Hg or a diastolic blood pressure ≥90 mm Hg or presence of antihypertensive drug treatment.

Eight-hour fasting blood samples were collected, and biological measurements were conducted in a Modular P apparatus (Roche Diagnostics, Basel, Switzerland) for the baseline and first follow-up, and in a Cobas 8000 (Roche Diagnostics, Basel, Switzerland) device for the second follow-up. The following analytical procedures [with maximum interbatch and intrabatch coefficients of variation (CVs)] were used: total cholesterol by CHOD-PAP (1.6%–1.7%) and high density lipoprotein (HDL)-cholesterol by CHOD-PAP+PEG+cyclodextrin (3.6%–0.9%). Glucose was assessed by glucose dehydrogenase (2.1%–1.0%) at baseline and by glucose hexokinase (1.6%–0.8%) at first and second follow-ups. Diabetes was defined as fasting plasma glucose ≥7.0 mmol/L or presence of an antidiabetic drug treatment.

## Statistical analysis

Statistical analyses were conducted using Stata v.16.1 (Stata Corp, College Station, TX, USA) separately for each survey. Results were expressed as number of participants (percentage) for categorical variables and as average±SD or median (IQR) for continuous variables. Bivariate comparisons between controlled and uncontrolled participants (using either ESC/EAS or GSLA criteria) were performed using chi-square for categorical variables and Student's t-test or Kruskal-Wallis nonparametric test for continuous variables. Multivariable analyses were conducted using logistic regression for categorical variables and results were expressed as multivariable-adjusted OR and 95% CI.

The associations between specific SNPs and management of dyslipidaemia were assessed by comparing the distribution of the genotypes according to controlled and uncontrolled participants (as defined by ESC/EAS or GSLA criteria) using Fisher's exact test.

Statistical significance was considered for a two-sided test with $p<0.05$.

## Patient and public involvement

None.

## RESULTS

### Prevalence of dyslipidaemia and changes in statin category

Overall, there were 709, 1056 and 1151 eligible participants at baseline, first and second follow-ups, respectively, of whom 92 (13.0%), 212 (20.1%) and 353 (30.7%) were excluded, leaving 617, 844 and 798 participants for analysis. The reasons for exclusion are indicated in online supplemental figure 1; the main reason was lack of genetic data. The number of participants treated for dyslipidaemia changed between surveys depending on the number of participants newly treated and the number of participants who dropped out. The characteristics of the included and the excluded participants are summarised in online supplemental table 6; excluded participants were less frequently born in Switzerland, while no other consistent difference was found.

The distribution of the different types of lipid-lowering treatments for the three surveys is provided in online supplemental figure 2, and of the statin generations and potency for the first and second follow-ups are provided in online supplemental figure 3. Statins represented the first type of hypolipidaemic drug, but their predominance decreased with time. Prevalence of first generation statins decreased and prevalence of third generation statins increased. Prevalence of low potency statins decreased and high potency statins increased. When posology was considered, statin potency was considerably reduced, but trends were similar (online supplemental figure 3). This decrease in potency was most marked for intermediate potency statins (online supplemental figure 4).

### Prevalence and factors associated with control of dyslipidaemia, ESC/EAS criteria

Prevalence of adequately managed dyslipidaemia was 52%, 45% and 46% at baseline, first and second follow-ups, respectively. The results of the analysis using the ESC/EAS criteria stratified by survey are summarised in tables 1–3.

On bivariate analysis (table 1), controlled participants were younger, had lower levels of cardiovascular risk factors and CVD risk and a higher prevalence of parental history of CVD than inadequately controlled participants in all surveys. Controlled participants also had a lower BMI and were taking less drugs than inadequately controlled participants in the first and second follow-ups; prevalence of fibrates was higher among inadequately controlled participants at baseline and in the first follow-up. No differences were found regarding GRSs between controlled and inadequately controlled participants in all surveys. On multivariable analysis (table 2), increased age or CVD risk was negatively associated with control in all surveys; no association was found between type of hypolipidaemic drug or quartiles of the LDL GRS and dyslipidaemia control.

The distribution of the statin generation or potency according to dyslipidaemia control is provided in online supplemental table 7. Controlled participants had a higher prevalence of third generation (first follow-up) or high potency statins than inadequately controlled participants. When posology was used to estimate potency, no differences were found. The results of the multivariable analyses taking into account statin generation or statin potency irrespective of the posology are provided in table 3 and online supplemental table 8, respectively. In both analyses, increasing age or CVD risk led to a lower likelihood of being controlled, while increasing statin generation or potency led to a higher likelihood of being controlled. When posology was used to estimate potency, the association was no longer significant (online supplemental table 9).

### Prevalence and factors associated with control of dyslipidaemia, GSLA criteria

Prevalence of adequately managed dyslipidaemia was 70%, 68% and 83% in the baseline, first and second follow-ups, respectively. The results of the analysis using the GSLA criteria stratified by survey are summarised in online supplemental table 10–15.

On bivariate analysis, controlled participants had lower CVD risk (all surveys), lower BMI (first and second follow-ups) and lower prevalence of smoking (first follow-up) than inadequately controlled participants; no differences were found regarding GRS (online supplemental table 10). On multivariable analysis, increased CVD risk was negatively associated with dyslipidaemia control in all surveys; men had a higher likelihood of being controlled (baseline and second follow-up) and alcohol consumption decreased likelihood of control (baseline); no association was found with LDL GRS or the

**Table 1** Bivariate comparison of socioeconomic and clinical characteristics among participants treated for dyslipidaemia, according to controlled and uncontrolled status as per European Society of Cardiology/European Atherosclerosis Society criteria

| | Baseline | | | First follow-up | | | Second follow-up | | |
|---|---|---|---|---|---|---|---|---|---|
| | Uncontrolled | Controlled | P value | Uncontrolled | Controlled | P value | Uncontrolled | Controlled | P value |
| N | 295 | 322 | | 465 | 379 | | 428 | 370 | |
| Age (years) | 64.0±8.4 | 59.5±8.0 | <0.001 | 67.4±8.6 | 61.0±7.7 | <0.001 | 71.4±8.2 | 64.4±8.9 | <0.001 |
| Women (%) | 118 (40.1) | 145 (44.9) | 0.233 | 186 (40.0) | 223 (58.8) | <0.001 | 218 (50.9) | 183 (49.5) | 0.678 |
| Swiss national (%) | 210 (71.4) | 220 (68.1) | 0.371 | 339 (72.9) | 261 (68.9) | 0.198 | 312 (72.9) | 251 (67.8) | 0.118 |
| Education (%) | | | 0.917 | | | 0.512 | | | 0.049 |
| High | 31 (10.5) | 35 (10.8) | | 62 (13.3) | 61 (16.1) | | 52 (12.2) | 68 (18.4) | |
| Middle | 61 (20.8) | 71 (22.0) | | 102 (21.9) | 83 (21.9) | | 99 (23.1) | 81 (21.9) | |
| Low | 202 (68.7) | 217 (67.2) | | 301 (64.7) | 235 (62.0) | | 277 (64.7) | 221 (59.7) | |
| Married/couple (%) | 199 (67.7) | 229 (70.9) | 0.388 | 271 (58.3) | 234 (61.7) | 0.308 | 233 (54.4) | 215 (58.1) | 0.298 |
| BMI (kg/m$^2$) | 28.2±4.4 | 27.8±4.6 | 0.217 | 28.1±4.6 | 27.1±4.9 | 0.002 | 28.1±4.6 | 27.0±5.1 | 0.002 |
| BMI categories (%) | | | 0.250 | | | 0.001 | | | 0.001 |
| Normal | 65 (22.1) | 90 (27.9) | | 111 (23.9) | 132 (34.8) | | 112 (26.2) | 142 (38.4) | |
| Overweight | 136 (46.3) | 141 (43.7) | | 216 (46.5) | 162 (42.7) | | 185 (43.2) | 142 (38.4) | |
| Obese | 93 (31.6) | 92 (28.5) | | 138 (29.7) | 85 (22.4) | | 131 (30.6) | 86 (23.2) | |
| Smoking (%) | | | 0.714 | | | 0.239 | | | 0.151 |
| Never | 101 (34.4) | 109 (33.8) | | 153 (32.9) | 142 (37.5) | | 168 (39.3) | 122 (33.0) | |
| Former | 117 (39.8) | 138 (42.7) | | 224 (48.2) | 161 (42.5) | | 190 (44.4) | 175 (47.3) | |
| Current | 76 (25.9) | 76 (23.5) | | 88 (18.9) | 76 (20.1) | | 70 (16.4) | 73 (19.7) | |
| Alcohol drinker (%) | 224 (76.2) | 232 (71.8) | 0.218 | 352 (75.7) | 280 (73.9) | 0.544 | 285 (73.3) | 240 (71.2) | 0.538 |
| Treatment for (%) | | | | | | | | | |
| Hypertension | 158 (53.7) | 145 (44.9) | 0.028 | 266 (57.2) | 170 (44.9) | <0.001 | 258 (60.3) | 167 (45.1) | <0.001 |
| Diabetes | 68 (23.1) | 27 (8.4) | <0.001 | 121 (26.0) | 27 (7.1) | <0.001 | 102 (23.8) | 55 (14.9) | 0.001 |
| Parental history (%) | 63 (21.4) | 101 (31.3) | 0.006 | 95 (20.4) | 123 (32.5) | <0.001 | 95 (22.2) | 116 (31.4) | 0.003 |
| CVD risk (%) | | | <0.001 | | | <0.001 | | | <0.001 |
| Other | 95 (32.3) | 173 (53.6) | | 140 (30.1) | 225 (59.4) | | 162 (37.9) | 193 (52.2) | |
| High | 66 (22.5) | 118 (36.5) | | 98 (21.1) | 107 (28.2) | | 73 (17.1) | 94 (25.4) | |
| Very high | 133 (45.2) | 32 (9.9) | | 227 (48.8) | 47 (12.4) | | 193 (45.1) | 83 (22.4) | |
| Number of drugs | | | | | | | | | |
| Including OTC | – | – | | 4 (3–6) | 4 (2–5) | <0.001* | 5 (3–7) | 4 (3–7) | <0.001* |
| Excluding OTC | – | – | | 4 (2–6) | 3 (2–5) | <0.001* | 4 (3–7) | 3 (2–6) | <0.001* |

Continued

**Table 1** Continued

| | Baseline | | | First follow-up | | | Second follow-up | | |
|---|---|---|---|---|---|---|---|---|---|
| | Uncontrolled | Controlled | P value | Uncontrolled | Controlled | P value | Uncontrolled | Controlled | P value |
| Genetic risk scores | | | | | | | | | |
| Total cholesterol | −2.8±9.4 | −3.9±9.7 | 0.149 | −3.5±9.2 | −3.6±8.9 | 0.941 | −4.2±9.6 | −3.2±8.1 | 0.147 |
| LDL–cholesterol | −2.2±7.8 | −3.2±7.2 | 0.117 | −2.5±7.5 | −2.3±7.1 | 0.827 | −3.2±7.6 | −2.5±6.3 | 0.171 |
| HDL–cholesterol | −6.5±3.5 | −6.8±3.4 | 0.215 | −6.6±3.6 | −6.9±3.6 | 0.371 | −6.5±3.6 | −6.9±3.6 | 0.124 |
| Hypolipidaemic drug treatment (%) | | | | | | | | | |
| Statins | 270 (91.5) | 308 (95.7) | 0.035 | 373 (80.4) | 288 (75.8) | 0.107 | 328 (76.6) | 264 (71.4) | 0.089 |
| Fibrates | 28 (9.5) | 17 (5.3) | 0.044 | 28 (6.0) | 5 (1.3) | <0.001 | 18 (4.2) | 14 (3.8) | 0.762 |
| Other | 11 (3.7) | 11 (3.4) | 0.834 | 72 (15.5) | 79 (20.8) | 0.047 | 84 (19.6) | 85 (23.0) | 0.249 |

Data from the baseline (2003–2006), first (2009–2012) and second (2014–2017) follow-ups of the CoLaus|PsyCoLaus study, Lausanne, Switzerland.
Results are expressed as number of participants (column %) for categorical variables and as average ± SD or as median (IQR) for continuous variables. Between-groups comparisons performed using $\chi^2$ for categorical variables and Student's t-test or Kruskal-Wallis nonparametric test (*) for continuous variables.
BMI, body mass index; CVD, cardiovascular disease; HDL, high density lipoproteins; LDL, low density lipoproteins; OTC, over the counter.

class of hypolipidaemic drug (online supplemental table 11).

The distribution of the statin generation or potency according to dyslipidaemia control is provided in online supplemental table 12. Controlled participants had a higher prevalence of third generation or high potency statins than inadequately controlled participants. When posology was used to estimate potency, no differences were found. The results of the multivariable analyses taking into account statin generation or statin potency irrespective of the posology are provided in online supplemental tables 13 and 14, respectively. In both analyses, increasing CVD risk led to a lower likelihood of being adequately controlled, while being male, increasing number of drugs or increasing statin generation or potency led a higher likelihood of being adequately controlled. When posology was used to estimate statin potency, the association was no longer significant (online supplemental table 15).

### Specific SNPs
The p values for the associations between 51 specific SNPs and dyslipidaemia control are presented in online supplemental table 16. Most statistically significant associations were found for *SLCO1B1* (Solute Carrier Organic Anion Transporter Family Member 1B1), but no consistent association was found overall.

## DISCUSSION
Our results show that individuals at high risk of CVD present an increased risk of mismanagement of dyslipidaemia, which was consistent among the three survey periods. The use of more potent statins increased the likelihood of dyslipidaemia control, while lipid GRSs were not associated with dyslipidaemia control.

### Prevalence of controlled dyslipidaemia
Prevalence of controlled dyslipidaemia varied between 45% and 52% according to ESC/EAS criteria and between 68% and 83% according to GSLA criteria. Those values are higher than the values reported by the EUROASPIRE IV, where a third (32.7%) of participants achieved the target level of <2.5 mmol/L for LDL-cholesterol.[16] Still, comparisons are difficult as EUROASPIRE IV focused on high-risk participants only. Importantly, prevalence rates of adequately managed dyslipidaemia were much higher using GSLA than ESC/EAS criteria. Hence, a clinician using GSLA criteria will lower LDL levels to a lesser level than using ESC/EAS criteria. Given that CVD risk decreases linearly with LDL-cholesterol levels (all other factors being equal),[17] the decrease in CVD risk is expected to be lower using GSLA than using ESC/EAS criteria. It would be important to evaluate if people managed according to the GSLA criteria achieve the same level of protection against CVD as if they were managed according to the ESC/EAS criteria.

### Factors associated with controlled dyslipidaemia
Participants at high risk of CVD had a higher likelihood of being inadequately managed in all surveys, irrespective

Table 2 Multivariable analysis of the factors associated with dyslipidaemia control as per European Society of Cardiology/European Atherosclerosis Society criteria

| | Baseline | | First follow-up | | Second follow-up | |
|---|---|---|---|---|---|---|
| | OR (95% CI) | P value | OR (95% CI) | P value | OR (95% CI) | P value |
| Age (per 10 years increase) | 0.44 (0.34 to 0.57) | <0.001 | 0.34 (0.27 to 0.43) | <0.001 | 0.35 (0.28 to 0.44) | <0.001 |
| Man vs woman | 0.89 (0.58 to 1.36) | 0.599 | 0.48 (0.34 to 0.70) | <0.001 | 1.23 (0.85 to 1.79) | 0.273 |
| Swiss vs Non-Swiss | 1.30 (0.85 to 1.97) | 0.223 | 1.13 (0.79 to 1.64) | 0.503 | 1.01 (0.69 to 1.48) | 0.948 |
| Education | | | | | | |
| High | 1 (ref.) | | 1 (ref.) | | 1 (ref.) | |
| Middle | 1.11 (0.54 to 2.27) | 0.782 | 0.83 (0.48 to 1.45) | 0.521 | 0.61 (0.35 to 1.07) | 0.087 |
| Low | 1.11 (0.59 to 2.11) | 0.738 | 0.81 (0.49 to 1.33) | 0.407 | 0.65 (0.40 to 1.07) | 0.093 |
| *P value for trend* | 0.738 | | 0.407 | | 0.093 | |
| Married vs not married | 1.21 (0.80 to 1.82) | 0.374 | 1.21 (0.86 to 1.71) | 0.270 | 1.06 (0.74 to 1.5) | 0.759 |
| Body mass index categories | | | | | | |
| Normal | 1 (ref.) | | 1 (ref.) | | 1 (ref.) | |
| Overweight | 0.59 (0.37 to 0.94) | 0.028 | 0.85 (0.57 to 1.26) | 0.413 | 0.61 (0.40 to 0.92) | 0.019 |
| Obese | 0.92 (0.54 to 1.56) | 0.753 | 0.96 (0.59 to 1.55) | 0.852 | 0.64 (0.40 to 1.04) | 0.074 |
| *P value for trend* | 0.753 | | 0.852 | | 0.074 | |
| Smoking categories | | | | | | |
| Never | 1 (ref.) | | 1 (ref.) | | 1 (ref.) | |
| Former | 1.54 (0.98 to 2.40) | 0.059 | 1.10 (0.75 to 1.62) | 0.618 | 1.38 (0.94 to 2.02) | 0.099 |
| Current | 0.63 (0.36 to 1.08) | 0.094 | 0.85 (0.52 to 1.39) | 0.512 | 1.04 (0.61 to 1.75) | 0.892 |
| *P value for trend* | 0.094 | | 0.512 | | 0.892 | |
| Alcohol drinker (yes vs no) | 0.66 (0.42 to 1.03) | 0.068 | 0.91 (0.62 to 1.35) | 0.645 | 0.64 (0.43 to 0.94) | 0.024 |
| Antihypertensive treatment (yes vs no) | 1.05 (0.70 to 1.57) | 0.820 | 1.18 (0.82 to 1.69) | 0.368 | 0.77 (0.53 to 1.12) | 0.176 |
| Parental history (yes vs no) | 1.00 (0.64 to 1.56) | 0.998 | 1.08 (0.74 to 1.59) | 0.690 | 0.79 (0.53 to 1.18) | 0.256 |
| CVD risk | | | | | | |
| Other | 1 (ref.) | | 1 (ref.) | | 1 (ref.) | |
| High | 1.25 (0.76 to 2.04) | 0.379 | 0.68 (0.45 to 1.04) | 0.074 | 1.42 (0.88 to 2.28) | 0.153 |
| Very high | 0.11 (0.06 to 0.18) | <0.001 | 0.12 (0.08 to 0.19) | <0.001 | 0.38 (0.25 to 0.59) | <0.001 |
| *P lue for trend* | <0.001 | | <0.001 | | <0.001 | |
| LDL genetic risk score quartiles | | | | | | |
| First | 1 (ref.) | | 1 (ref.) | | 1 (ref.) | |
| Second | 1.11 (0.65 to 1.89) | 0.696 | 0.97 (0.61 to 1.54) | 0.883 | 1.31 (0.81 to 2.12) | 0.264 |
| Third | 0.70 (0.41 to 1.19) | 0.185 | 0.96 (0.60 to 1.53) | 0.866 | 1.57 (0.97 to 2.51) | 0.064 |
| Fourth | 0.74 (0.43 to 1.25) | 0.259 | 1.07 (0.67 to 1.72) | 0.764 | 1.44 (0.89 to 2.32) | 0.133 |
| *P value for trend* | 0.107 | | 0.781 | | 0.099 | |
| Hypolipidaemic drug treatment | | | | | | |
| Statins | 1.56 (0.18 to 13.8) | 0.690 | 1.00 (0.55 to 1.81) | 0.998 | 1.42 (0.81 to 2.51) | 0.223 |
| Fibrates | 0.67 (0.08 to 5.35) | 0.707 | 0.13 (0.04 to 0.43) | 0.001 | 1.20 (0.46 to 3.15) | 0.714 |
| Other | 0.94 (0.34 to 2.61) | 0.910 | 0.57 (0.31 to 1.04) | 0.067 | 0.71 (0.40 to 1.26) | 0.237 |

Data from the baseline (2003–2006), first (2009–2012) and second (2014–2017) follow-ups of the CoLaus|PsyCoLaus study, Lausanne, Switzerland. Results are expressed as odds ratio and (95% CI). Statistical analysis wase done using logistic regression.
CVD, cardiovascular disease.

of the criteria considered. Those findings are consistent with a recent review of European studies[18] and a large cross-sectional European study,[19] where control rates were lower than 20%. Possible explanations include the fact that subjects at high CVD risk should have much lower lipid values, thus more difficult to achieve.

Similar to a Polish study[11] but contrary to a US study,[20] the prevalence of highly potent statins increased from

**Table 3** Multivariable analysis of the factors associated with dyslipidaemia control as per European Society of Cardiology/European Atherosclerosis Society criteria

| | First follow-up | | Second follow-up | |
|---|---|---|---|---|
| | OR (95% CI) | P value | OR (95% CI) | P value |
| Age (per 10 years increase) | 0.31 (0.23 to 0.41) | <0.001 | 0.35 (0.27 to 0.47) | <0.001 |
| Man vs woman | 0.60 (0.39 to 0.92) | 0.018 | 1.40 (0.90 to 2.16) | 0.134 |
| Swiss vs Non-Swiss | 1.17 (0.76 to 1.80) | 0.474 | 1.16 (0.75 to 1.80) | 0.502 |
| Education | | | | |
| High | 1 (ref.) | | 1 (ref.) | |
| Middle | 0.78 (0.40 to 1.53) | 0.476 | 1.08 (0.54 to 2.15) | 0.821 |
| Low | 0.88 (0.49 to 1.59) | 0.681 | 1.04 (0.57 to 1.90) | 0.895 |
| *P value for trend* | 0.681 | | 0.895 | |
| Married vs not married | 1.23 (0.82 to 1.83) | 0.317 | 1.19 (0.79 to 1.78) | 0.407 |
| Body mass index categories | | | | |
| Normal | 1 (ref.) | | 1 (ref.) | |
| Overweight | 0.84 (0.52 to 1.35) | 0.474 | 0.56 (0.35 to 0.92) | 0.023 |
| Obese | 0.91 (0.53 to 1.58) | 0.749 | 0.54 (0.31 to 0.95) | 0.032 |
| *P value for trend* | 0.749 | | 0.032 | |
| Smoking categories | | | | |
| Never | 1 (ref.) | | 1 (ref.) | |
| Former | 1.09 (0.70 to 1.71) | 0.695 | 1.21 (0.78 to 1.88) | 0.384 |
| Current | 0.84 (0.46 to 1.51) | 0.560 | 0.93 (0.51 to 1.71) | 0.820 |
| *P value for trend* | 0.560 | | 0.820 | |
| Alcohol drinker (yes vs no) | 0.79 (0.50 to 1.25) | 0.316 | 0.72 (0.46 to 1.12) | 0.146 |
| AntiHTA ttt (yes vs no) | 0.97 (0.63 to 1.51) | 0.903 | 0.80 (0.52 to 1.25) | 0.337 |
| Parental history (yes vs no) | 1.27 (0.80 to 2.02) | 0.310 | 0.76 (0.48 to 1.23) | 0.267 |
| CVD risk | | | | |
| Other | 1 (ref.) | | 1 (ref.) | |
| High | 0.63 (0.39 to 1.02) | 0.061 | 1.32 (0.76 to 2.31) | 0.327 |
| Very high | 0.08 (0.05 to 0.14) | <0.001 | 0.35 (0.21 to 0.58) | <0.001 |
| *P value for trend* | <0.001 | | <0.001 | |
| LDL genetic risk score quartiles | | | | |
| First | 1 (ref.) | | 1 (ref.) | |
| Second | 0.90 (0.53 to 1.54) | 0.707 | 1.67 (0.95 to 2.93) | 0.076 |
| Third | 0.89 (0.52 to 1.52) | 0.665 | 1.79 (1.04 to 3.07) | 0.036 |
| Fourth | 1.11 (0.65 to 1.92) | 0.696 | 1.64 (0.93 to 2.86) | 0.085 |
| *P value for trend* | 0.725 | | 0.085 | |
| Number of drugs (per one unit) | 1.15 (1.05 to 1.25) | 0.002 | 1.07 (0.99 to 1.15) | 0.069 |
| Statin generation | | | | |
| First | 1 (ref.) | | 1 (ref.) | |
| Second | 1.90 (1.18 to 3.05) | 0.008 | 1.90 (1.08 to 3.36) | 0.026 |
| Third | 3.62 (1.65 to 7.92) | 0.001 | 2.18 (1.05 to 4.51) | 0.036 |
| *P value for trend* | 0.001 | | 0.036 | |
| Fibrates | NC | | 2.55 (0.19 to 34.1) | 0.480 |
| Other hypolipidaemic drugs | 0.90 (0.38 to 2.11) | 0.800 | 1.13 (0.51 to 2.51) | 0.762 |

Data from the first (2009–2012) and second (2014–2017) follow-ups of the CoLaus|PsyCoLaus study, Lausanne, Switzerland. Analysis was done taking into account statin generation.
Results are expressed as OR and (95% CI). Statistical analysis was done using logistic regression.
NC, not computable; antiHTA ttt, antihypertensive drug treatment.

41% in 2009–2012 to 56% in 2014–2017. This value is higher than reported in the EUROASPIRE V[4] study, where 49.9% of participants were on high-intensity therapy. Importantly, participants on high potency statins achieved better control, a finding also reported in the EUROASPIRE V[4] and the DA VINCI[19] studies. The 2019 ESC/EAS guidelines for management of dyslipidaemia recommend that high potency statins at highest recommended tolerable dose be initially applied to control lipid levels.[3] Our results thus strengthen the importance of such recommendation, and general practitioners should be urged to shift to more potent statins to achieve better results. Still, our results also suggest that, despite a higher prescription rate of highly potent statins, those drugs are not prescribed at their full potency/posology, and that a sizeable fraction of treated subjects fails to reach lipid targets.

Other clinical and sociodemographic factors were associated with dyslipidaemia control, but associations were inconsistent between study periods or between ESC/EAS and GSLA criteria. Increasing age was negatively associated with dyslipidaemia control using ESC/EAS criteria but no association was found using GSLA criteria. Either no association,[21] an inverse association[22] or a positive association[23] between age and dyslipidaemia control have been reported. Similarly, men achieved better control than women using the GSLA criteria, while no consistent association was found using the ESC/EAS criteria. Better control rates have also been reported in the USA,[24] while the inverse association was reported in Germany.[22] Such discrepancies might be related to the criteria applied, as age and gender might be stronger or weaker determinants of CVD risk in some risk equations compared with others. No association was found between nationality, education, marital status, job type or BMI categories and dyslipidaemia control. Our findings replicate those of other studies where no association between education,[21] marital status[21 22] and dyslipidaemia control was found. Overall, our results suggest that the sociodemographic factors associated with dyslipidaemia control differ according to country and to the criteria used to estimate CVD risk.

### Genetic scores and individual SNPs

Several authors suggested that genetic profiling could be used to guide statin treatment and thus improve outcomes.[7 8] A meta-analysis published in 2015 concluded that people with the highest burden of genetic risk derived the largest relative and absolute clinical benefit from statin therapy,[25] although such statement could also apply to people for whom cardiovascular risk was assessed using clinical data. Further, the initial promises regarding genetic testing of the kinesin-like protein 6 (KIF6) gene to guide statin prescription (the StatinCheck test) were not confirmed.[26] In this study, no association between genetic scores for lipid markers and statin efficiency was found. Possible reasons include the small effect of each individual SNP,[27] as a set of 95 SNPs explained <15% of

total lipid variance,[28] or the progressive blunting of the genetic effect by advanced ageing as found for BP.[29] Thus, our results suggest that genetic profiling of subjects prior to initiation of statin therapy might be clinically irrelevant, and such profiling is not stated in the current ESC/EAS guidelines for the management of dyslipidaemias.[3] Nevertheless, several associations were found with the *SLCO1B1* gene. Some authors have suggested that genetic variations in this gene are associated with response to statins.[8] Hence, this gene might be of interest to adapt statin treatment, and it would be important that other studies be conducted to confirm our findings.

### Study limitations

This study has several limitations worth acknowledging. First, the sample size was relatively small and our study was likely underpowered to detect the minute associations between the genetic scores and dyslipidaemia control. Still, should those GRSs be applied in clinical practice, their effect should be large enough to allow choosing between several statins in a given individual. Second, the analysis was restricted to Switzerland, and findings might be generalisable to other countries or ethnicities. Still, most findings agree with larger studies such as EUROASPIRE V[4] or DA VINCI.[19] Third, there is no consensus regarding the management of dyslipidaemia, as thresholds for treatment vary according to country or scientific society.[30] Hence, our results cannot be extrapolated to other settings, and it would be important that similar studies be conducted in other countries.

### CONCLUSION

Management of dyslipidaemia is suboptimal in Switzerland, especially for individuals at high cardiovascular risk. The effectiveness of high potency statins is hampered by low posology. GRSs are not associated with dyslipidaemia control, but the effect of *SLCO1B1* in statin therapy should be further investigated.

**Contributors** VC: investigation, methodology, writing—original draft preparation, visualisation. PM-V: conceptualisation, methodology, data curation, formal analysis, writing—reviewing and editing, visualisation. NA: writing—reviewing and editing. JV: reviewing and editing. The authors had full access to the data and took responsibility for its integrity. All authors have read and agreed to the written manuscript. PM-V had full access to the data and is the guarantor of the study.

**Funding** The CoLaus|PsyCoLaus study was and is supported by research grants from GlaxoSmithKline (N/A), the Faculty of Biology and Medicine of Lausanne (N/A), and the Swiss National Science Foundation (grants 33CSCO-122661, 33CS30-139468, 33CS30-148401 and 33CS30_177535/1). ValeriyaChekanova received an excellence scholarship of the Swiss government (N/A) to conduct research in Switzerland for one year.

**Competing interests** None declared.

**Patient and public involvement** Patients and/or the public were not involved in the design, or conduct, or reporting or dissemination plans of this research.

**Patient consent for publication** Not applicable.

**Ethics approval** The institutional Ethics Committee of the University of Lausanne, which afterwards became the Ethics Commission of Canton Vaud (www.cer-vd.ch), approved the baseline (reference 16/03), the first (reference 33/09) and the second

(reference 26/14) follow-ups. The approval was confirmed in 2021 (reference PB_2018-00038, 239/09). The study was performed in agreement with the Helsinki declaration and its former amendments, and in accordance with the applicable Swiss legislation. All participants gave their signed informed consent before entering the study. Participants gave informed consent to participate in the study before taking part.

**Provenance and peer review** Not commissioned; externally peer reviewed.

**Data availability statement** Data are available upon reasonable request. Non-identifiable individual-level data are available for researchers who seek to answer questions related to health and disease in the context of research projects who meet the criteria for data sharing by research committees. Please follow the instructions at https://www.colaus-psycolaus.ch/ for information on how to submit an application for gaining access to CoLaus|PsyCoLaus data.

**ORCID iD**
Pedro Marques-Vidal http://orcid.org/0000-0002-4548-8500

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
