## [Reviewer comments · BMJ Open]

ARTICLE DETAILS

TITLE (PROVISIONAL)	Association of clinical and genetic risk factors with management of dyslipidemia: analysis of repeated cross-sectional studies in the general population of Lausanne, Switzerland
AUTHORS	Chekanova, Valeriya; Abolhassani, Nazanin; Vaucher, Julien; Marques-Vidal, Pedro

VERSION 1 – REVIEW

REVIEWER	Klop, Boudewijn
REVIEW RETURNED	15-Jul-2022

GENERAL COMMENTS	This is an interesting population study, which addresses the problem of under treatment of hyperlipidemia in people with increased cardiovascular risk. Insufficient treatment of hyperlipidemia is due to the simple fact of insufficient dosage and number of lipid lowering drugs and not due to genetic factors. Therefore, this study is relevant to all clinicians treating hyperlipidemia. However, I have would like to address some concerns with study. Major comments: 1. The important data are difficult to extract from the manuscript due to the large number of analyses and number of (supplemental) tables: statin generation, with or without posology, national versus international guideline etc.). It would be better to only include the most relevant analyses and tables to improve readability. In addition, the total number of study subjects is relatively limited to allow robust statistical analysis in all those variations and included variables.2. The cohort study design is unclear to me. How is it possible that the number of participants increased with follow-up. Were more study subjects from the same study population eligible for analysis during follow-up? Or was there every 5 years a new number of subjects from the Lausanne population? If the latter was the case it should not be described as a cohort study but as a prospective cross sectional study. Please clarify.3. The results can not readily be extrapolated to the European or Western situation due to a lack of consensus in use of guidelines (national versus international). Please discuss in which way this may have affected the results and interpretation of the data. Minor comments: 1. The abbreviation GRS should be clarified in the abstract.2. How large was the total Lausanne study population. Please mention in the manuscript.3. There is a typo on page 8, line 49.
---

	4. The differences between the Tables and the respective analysis they represent is insufficiently clarified in the Table descriptions. This should be improved.
REVIEWER	Ghayour Mobarhan, Majid Faculty of Medicine, Biochemistry&Nutrition
REVIEW RETURNED	12-Aug-2022
GENERAL COMMENTS	dear author, please rewrite abstract with detail in method section. beat regard

VERSION 1 – AUTHOR RESPONSE

Answers to Reviewer: 1 (Boudewijn Klop)

This is an interesting population study, which addresses the problem of under treatment of hyperlipidemia in people with increased cardiovascular risk. Insufficient treatment of hyperlipidemia is due to the simple fact of insufficient dosage and number of lipid lowering drugs and not due to genetic factors. Therefore, this study is relevant to all clinicians treating hyperlipidemia.

Major comments:

1. The important data are difficult to extract from the manuscript due to the large number of analyses and number of (supplemental) tables: statin generation, with or without posology, national versus international guideline etc.). It would be better to only include the most relevant analyses and tables to improve readability. In addition, the total number of study subjects is relatively limited to allow robust statistical analysis in all those variations and included variables.

We agree with the reviewer and we moved tables 4 and 5 to the supplementary information. We decided to keep all the analyses as we consider them important for both Swiss and International readers, and because they add extra arguments against using GRS in clinical practice.

2. The cohort study design is unclear to me. How is it possible that the number of participants increased with follow-up. Were more study subjects from the same study population eligible for analysis during follow-up? Or was there every 5 years a new number of subjects from the Lausanne population? If the latter was the case it should not be described as a cohort study but as a prospective cross sectional study. Please clarify.

Within each follow-up, we considered all participants treated for dyslipidemia. Hence, with time, the number of participants treated changed, depending on the number of participants newly treated and the number of treated participants who dropped out. We added the following statement in the results “The number of participants treated for dyslipidemia changed between surveys depending on the number of participants newly treated and the number of participants who dropped out”

3. The results can not readily be extrapolated to the European or Western situation due to a lack of consensus in use of guidelines (national versus international). Please discuss in which way this may have affected the results and interpretation of the data.

We agree with the reviewer and we added the following statement in the “strengths and limitations” bullet points:

- Lack of consensus regarding diagnosis and management of dyslipidemia; results cannot be extrapolated to other settings and populations
- Results based on a single population and hence not forcefully generalizable to other settings and populations

We also added the following statement in the “strengths and limitations” chapter in the main text: “Thirdly, there is no consensus regarding the management of dyslipidemia, as thresholds for treatment vary according to country or scientific society ³⁰. Hence, our results cannot be extrapolated to other settings, and it would be important that similar studies be conducted in other countries.”

Minor comments:

1. The abbreviation GRS should be clarified in the abstract.

The term “Genetic risk scores (GRS)” is now mentioned in the abstract. The editor also mentioned this issue

2. How large was the total Lausanne study population. Please mention in the manuscript.

The population size of Lausanne was 137,810 in 2017. This is now indicated in the methods, study population

3. There is a typo on page 8, line 49.

Corrected. Thank you for spotting it

4. The differences between the Tables and the respective analysis they represent is insufficiently clarified in the Table descriptions. This should be improved.

We completely revised the tables’ titles and tried to make them clearer and more informative

Answers to Reviewer: 2 (Dr. Majid Ghayour Mobarhan, Faculty of Medicine)

Please rewrite abstract with detail in method section.

The abstract has been rewritten taking into account the journal’s guidelines. As there were no interventions, this paragraph was omitted as per journal’s guidelines. The editor also commented on this issue.

VERSION 2 – REVIEW

REVIEWER	Klop, Boudewijn
REVIEW RETURNED	06-Feb-2023
GENERAL COMMENTS	This is an interesting study and the comments of the reviewers have been correctly addressed in this revised manuscript.